# Impact of the Minimum Head on Low-Head Hydropower Plants Energy Production and Profitability

**Bartosz Ceran [1],\* , Jakub Jurasz [2] , Robert Wróblewski [1], Adam Guderski [1], Daria Złotecka [1] and Łukasz Kaźmierczak [1]**

[1] Institute of Electrical Power Engineering, Poznan University of Technology, Piotrowo 3A, 60-965 Poznan, Poland; robert.wroblewski@put.poznan.pl (R.W.); adam.guderski@put.poznan.pl (A.G.); daria.zlotecka@put.poznan.pl (D.Z.); lukasz.kazmierczak@put.poznan.pl (Ł.K.)

[2] Department of Engineering Management, Faculty of Management, AGH University of Science and Technology, 30-059 Kraków, Poland; jjurasz@zarz.agh.edu.pl

\* Correspondence: bartosz.ceran@put.poznan.pl; Tel.: +48-61-665-2523

**Abstract:** In Poland, existing barrages are characterized by relatively high flow and low head, which is challenging for the effective utilization of theoretical watercourse power. The paper presents the impact of the minimum head of the hydro sets on the annual electricity production of small hydropower plants at low-head locations for two types of water turbines: Archimedes and Kaplan turbines. A developed mathematical model was used to simulate energy yield from Archimedes and Kaplan turbines for a given value of the minimum technical head, depending on the number of installed hydro sets. For economic analysis purposes, the levelized cost of electricity (LCOE) and net present value (NPV) indicators were calculated. The conducted research allowed for comparing Archimedes and Kaplan's turbine operating conditions and how the minimum head parameter influences their electricity production and utilization time. As concluded in the results, the influence of minimum head in energy production is more distinct for the Archimedes screw technology than for the Kaplan turbine. The research shows that the decrease in energy production associated with the hydro unit's minimum head parameter is from 0% to 30% for Kaplan, and it is 6% to 52% for Archimedes turbines.

**Keywords:** small-scale hydropower; Archimedes turbine; Kaplan turbine; minimum head; energy yield

## 1. Introduction

The Polish power system's generation sector is 70% based on coal-fired power plants [1]. Therefore, the production of 1 kWh of electricity corresponds to the emission of approximately 0.77 kg of carbon dioxide. Increasing environmental restrictions necessitate a transformation of the power sector structure in Poland to increase the share of renewable energy sources in the electricity production process [2]. According to data presented by the Polish transmission system operator (TSO), 5% of the installed capacity in the system are industrial hydropower plants. In comparison, 16% are wind farms and other renewable technologies, including dynamically developing photovoltaics. There is also a massive increase in units operating on gas fuel [3].

In Poland, Europe and worldwide, it is possible to increase the hydropower potential. Hoes et al. [4] estimated the world's gross theoretical potential of hydropower at 52 PWh/year. Small hydropower plants represent a significant part of this potential. In this category, particular attention should be paid

to small-scale hydropower plants (SHP) operating in low head conditions (below 5 m [5]), the significant potential of which is still not fully recognized.

Bódis et al. [6], estimating hydropower resources in Europe, excluded locations with heads below 6 m, which could be utilized in power generation using the currently available technology. Wiemann et al. [7] drew attention to the possibility of managing low heads (from 0.8 m to 2.0 m) in Europe, with the use of hydropower plants with a power of 100–1000 kW.

The technical potential of hydropower sources in Poland is estimated at 12–14 TWh per year [8]. This represents about 8% of the annual electricity consumption, which amounted to 170 TWh [3] in 2019. The current use of the aforementioned technical potential is estimated at 12%, a large part of SHPs built next to the existing hydraulic structures (e.g., weirs, dams). According to Punys et al. [9], Poland is the second European country, after France, in terms of the number of hydraulic structures where SHPs can be developed. The estimated total potential of hydropower generation of SHPs in Poland is approximately 1300 GWh/year [9].

Research on the practical use of water potential is the subject of many scientific publications. Zhou et al. [4] included the currently most critical types of turbines in the low head category: propeller turbines, tubular turbines, cross-flow Banki-Michell turbines, open flume Francis turbines, Kaplan turbines, Archimedes screws, and waterwheels. In the research, the last three technologies were given particular attention. Balkhair et al. [5] considered places with an available head below 5 m as low-head locations, while Zhou et al. [10] distinguished the category of even lower heads—from 3 m to 0 m (ultra-low-head). Lavric et al. [11] stated that locations with such small heads often remain unused. Only the development of low-head technologies permits their practical use.

In recent years, renewed interest in the already mature technology of waterwheels can be observed, which is reflected in a growing amount of publications on this subject. In [12], the author presented the design of a waterwheel where the maximum power output occurs at a certain wheel speed. In work [13], the authors tested the waterwheel efficiency in run-of-river conditions.

In [14], the authors optimized gravity waterwheels. The results showed that the maximum efficiency of overshot and undershot waterwheels was around 85%. In comparison, that of breastshot waterwheels ranged from 75% to 80%, depending on inflow configuration. The aim of the numerical simulations presented in [15] was to show the influence of the immersion depth value on the waterwheel characteristics. In [16], the authors tested the water inflow regulation to undershot waterwheel blades by changing the shape of the end part of the inlet channel. This action is intended to maximize the waterwheel's efficiency at low and variable flow rates.

Hydroelectric power plants based on Archimedes screws [17] began to be used around the year 2000, and they soon became a popular solution for low-head locations. Rohmer et al. [18] conducted the modeling and selecting of the turbine size for the planned flow. The authors examined the effect of excessive filling of the Archimedes screw trough on the occurring energy losses. They compared the theoretical results with experimental results. However, the authors of the work did not mention the occurrence of a minimum head, which is essential for the topic under consideration.

In work [19], the authors examined the influence of the angle of inclination of the Archimedes screw turbine on its energy efficiency. Similar to the previously mentioned work, they indicated losses caused by a high level of filling the turbine trough. Moreover, the study distinguishes two optimal inclination angles of the hydro unit, including the local maxima of efficiency, occurring at two different head values. However, as can be deduced from the study, the authors did not investigate the hydro unit's operation under the conditions of a variable head for the previously specified inclination angle of the Archimedes screw. Such conditions occur in real-life objects. In [20], the authors presented a method for measuring the hydroelectric trough filling height to investigate the overflow of working screws as a function of the trough filling height. The studies did not take into account the variability of the head.

On the other hand, the authors of work [21] analyzed the influence of the blade angle and the number of turbine blades on Archimedes hydro unit efficiency, taking into account head changes.

Moreover, the authors investigated the usefulness of CFD modeling in the implementation of simulation tests. This method brings very reliable results under the conditions mentioned. A specific aspect of these studies was the determining of a different optimal operating angle for turbines with a different number of blades.

The authors of [22] described the negative impact of the decreasing head on the efficiency value and the range of achieved rotational speeds of the turbine. The head is adjusted by the level of lower reservoir changes. The paper presents the experimental results of research studies, which indicated a rapid decrease in the hydro unit's efficiency when increasing the lower reservoir level. Not only does the peak point of the efficiency curve drop, but the curve itself changes shape. What is essential for the analyses is that the authors show the negative impact of the increase in the lower reservoir level on the hydropower plant's operation.

In [23], the authors investigated the efficiency of an Archimedes turbine with a different number of blades using computer simulations in the ANSYS environment. On the other hand, in [24], the authors proved that the average flow and electricity generated by an Archimedes turbine are inversely correlated. The turbine they tested working on a variable flow produces 2% less energy than a constant flow turbine.

One of the most efficient technologies in low-head hydropower plants is a tubular turbine with a Kaplan turbine rotor [25]. A large number of scientific publications focus on this type of hydroelectric units. Mulu et al. [26] focused on the experimental studies of flows in the tubular turbine at the point of the highest efficiency and their influence on turbine components. Abeykoon et al. [27] researched the influence of the shape of a Kaplan turbine runner wheel on the hydro unit efficiency. After theoretical calculations, the structure was optimized by simulation in the ANSYS environment, which improved the hydro unit's efficiency. Abbas et al. [28] focused on the optimal design of a turbine to be used in wastewater treatment plants. According to the manufacturer [29], tubular Kaplan turbines can be currently used from 1.5 m head.

Low-head technologies have gained high popularity in recent years, as confirmed by the above-mentioned publications. However, research into low-head technologies focuses mainly on optimizing the design parameters or maximizing the efficiency of converting primary energy into electricity. Only Lavric et al. [11,17,24] considered in their research the shutdown of hydro sets forced by the flow and the concurrent head, showing temporary shutdowns.

In spite of very valuable analysis, the authors of these publications do not sufficiently address the impact of the technical minimums of the effective head on the annual production of electricity. Determination of the amount of energy lost due to hydro set shutdowns serves as the starting point for the reduction of such losses. It may inspire numerous scientific research projects to focus on, for example, the analysis of structural alterations in turbines or on modifications of turbine selection procedure at the stage of power plant design.

Failure to take into account energy losses resulting from hydro set shutdowns and the lack of research into the impact of the minimum technical value on the production of energy has become a gap in scientific research on this subject matter. Addressing this topic, carrying out calculations leading to the determination of energy lost in this manner, as well as demonstrating the significance of this phenomenon is an innovation presented in this article, and it is intended to bridge the above-mentioned gap.

The primary purpose of this study is to investigate the impact of the minimum head of the considered hydro sets on the annual electricity production of small hydropower plants at low-head locations. Archimedes turbines and tubular turbines with Kaplan rotors were used in the case study. Such analyses have not been carried out for Poland so far. The arguments for the selection of the above-mentioned two technologies are the nominal parameters of the considered barrage, falling in the range of application of these technologies [10] and the effectiveness of energy conversion in the range of the analyzed hydrological conditions [11].

## 2. Problem Description

The first of the considered technologies is the Archimedes turbine. Scientific research does not provide any explicit areas of application for this technology. Elbatran et al. [30] stated that its use was possible from 1 m head, Zhou et al. [10] considered 10 m as the upper limit, and Rohmer et al. [9] gave the overall applicable head as 1–6.5 m.

The efficiency of water energy conversion into mechanical energy received on the shaft is estimated at 80%, and the tendency increases together with increasing turbine diameter [31]. However, the studies in [32] give the highest efficiency value at 89%. The authors quote 4 m as the largest possible Archimedes screw diameter. According to the authors, this value is related to the problem of cracking welds, which connect the shaft with the spiral blades. The initial condition torque values come from the developed Archimedes turbine model with an assumed water flow rate from 0 to 2.7 $m^3$/s [33]; while in [34], the authors assumed the water flow rate from 0.25 to 6.5 $m^3$/s.

The second technology under consideration is a tubular turbine with a Kaplan rotor, one of the Kaplan turbines [35] dedicated to low-head locations. It is a solution with a much more extended period of application in the hydropower industry; however, research on its improvement is still carried out [26]. Turbines with unit power of up to 1 MW made in this technology are mainly used for heads of 1–17 m, while hydro-sets with a higher rated power are applied for higher heads [36]. The efficiency of Kaplan turbines depends on the properties of a given unit [37]. The research in [25] showed that tubular turbine efficiency with Kaplan rotors might exceed 90% and even reach 91.7%. The diameters of classic Kaplan turbines used in the world exceed 7 m. Units with such dimensions have a unit flow of 400 $m^3$/s. Simultaneously, this technology is also suitable for much smaller flows, even in the range of 0–1.5 $m^3$/s [38].

As a case study, the location shown in Figure 1 was adopted, with a nominal head of 1.6 m. The exact coordinates are latitude 52.8648 and longitude 16.0676. This specific location is at the 170th kilometer of the Noteć River (total length of 388 km, and a catchment area of 17,330 $km^2$).

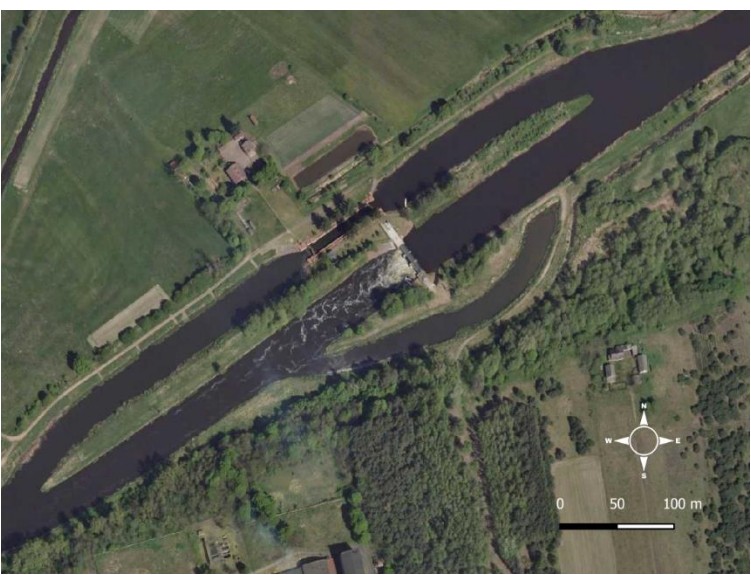

**Figure 1.** Location of the case study. Source: Open Street Map.

There is a hydrological station at the adjacent barrage, measuring the flow and water level on the barrage. There are no significant tributaries between these two stages, which could significantly affect the water flow variation passing through the two stages. Therefore, these flow values will be considered the same for the step under consideration in the following part of this paper. At the Krzyż barrage, the following characteristic flows [39] occur:

- average low flow—26 $m^3$/s (SNQ);

- average flow from the average annual flows—54 m$^3$/s (SSQ);
- average high flow—97 m$^3$/s (SWQ).

There are several similar barrages along the Noteć River, some of which have been developed for power generation [40]. Płaczek et al. [41] distinguish four sections with different terrain characteristics. The location under consideration is placed within the section known as the Canalized Noteć.

## 3. Model and Method Description

### 3.1. Estimating Energy Production—Small Hydropower Plant Model

According to [42], the nominal head of the Drawsko barrage is 1.6 m. The Manning formula was used for uniform flows in open flumes (1). This formula will be used later to determine the estimated head-flow relation.

$$Q = I^{\frac{1}{2}} \cdot A \cdot R_h^{\frac{2}{3}} \cdot n^{-1}, \tag{1}$$

where: Q—water flow [m$^3$/s], I—hydraulic gradient [‰], Rh—hydraulic radius [m], n—trough roughness coefficient [s/m1/3], A—cross-sectional area [m$^2$],

The water flume cross-section area (Figure 2) is defined by Formula (2):

$$A = (b + mh) \cdot h, \tag{2}$$

where: h—filling of the trough [m], b—width of the trough bottom [m], m—cotangent of inclination angle α[-] (α—the slope of the river trough)

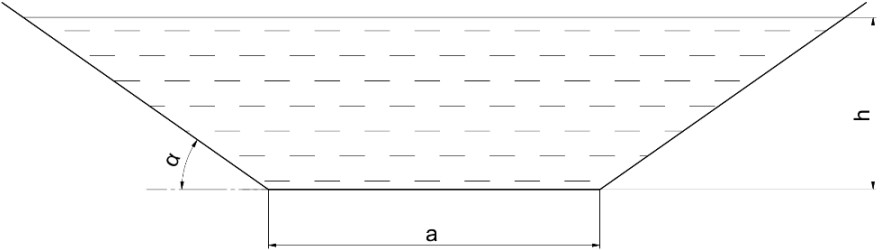

**Figure 2.** Cross-section of the Noteć river trough model below the Drawsko barrage (own study).

Then it is necessary to calculate the value of the hydraulic radius Rh according to Formula (3):

$$R_h = \frac{(b + mh)h}{b + 2h\sqrt{1 + m^2}}, \tag{3}$$

After substituting Formulas (2) and (3) to Formula (1), the relationship Q(h) is obtained as described by Formula (4).

$$Q(h) = I^{\frac{1}{2}} \cdot (b + mh) \cdot h \cdot \left( \frac{(b + mh)h}{b + 2h\sqrt{1 + m^2}} \right)^{\frac{2}{3}} \cdot n^{-1} \tag{4}$$

The data adopted for the model are summarized in Table 1. Their accuracy and reliability are crucial for the credibility of the model. The critical parameter that undergoes the most dynamic changes in the roughness coefficient of the river trough. According to [43], its value depends on, among other things, the state of riverbed vegetation, which varies together with seasonal changes. The b parameter was determined based on satellite images.

**Table 1.** Summary of data adopted for the presented calculation methodology [41].

| Parameter | Data | Unit |
|-----------|------|------|
| n | 0.035 | s/m1/3 |
| I | 0.002 | - |
| b | 29.2 | m |

According to the above model, a curve determining the flume's filling as a function of flow was obtained, which is shown in Figure 3. It allowed determining the head-flow relation, assuming a constant upper reservoir level. The assumption of a constant upper reservoir level results from the barrage construction and the maintenance of the required shipping parameters on the waterway. The adopted water level in the upper reservoir is directly related to the issued water permit requirements for the analyzed barrage. The assumed water level in the upper reservoir results from the necessity to maintain the shipping conditions.

The data presented in Figure 3 were used to calculate the head as a function of flow. The barrage head value at zero flow was obtained by calibrating the model to flow at a nominal head. Nominal head means the difference of water levels on the upper and lower reservoirs at the flow equal to the average of the lowest annual flows (SNQ). For the examined location, it is 26 $m^3$/s [39].

Figure 4 compares the model data with empirical data. The model has a mean square error (MSE) of 0.041, and in the flow range of 50–95 $m^3$/s, it is 0.0246. When it comes to the flow of 50 $m^3$/s, the value of MSE is less favorable, reaching an error value of 0.065, which will be discussed in the following paragraphs.

The head-flow model takes into account flows in the range of 10–150 $m^3$/s. This is due to the minimum (13.2 $m^3$/s), and maximum (149 $m^3$/s) flow recorded at the adjacent barrage over the last 50 years [44]. The applied model has two simplifying assumptions. The first is to assume the flow below the considered stage as a uniform flow (water movement parameters are constant along the length of the considered river trough). It is a slowly changing flow (water movement parameters are variable). In the case of hydraulic support by the step below, the model head value will be overestimated compared to the actual values at low flows. The second simplification is the assumption of a constant slope of the river bed; however, it is difficult to indicate the resulting negative impact of the model.

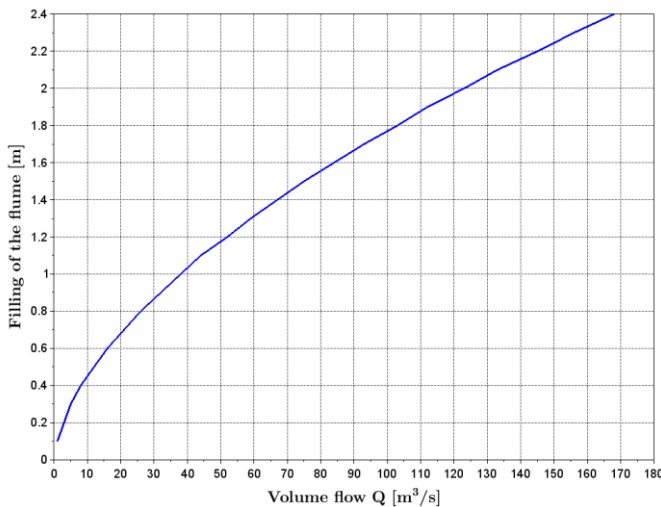

**Figure 3.** Flume filling model as a function of the current volume flow.

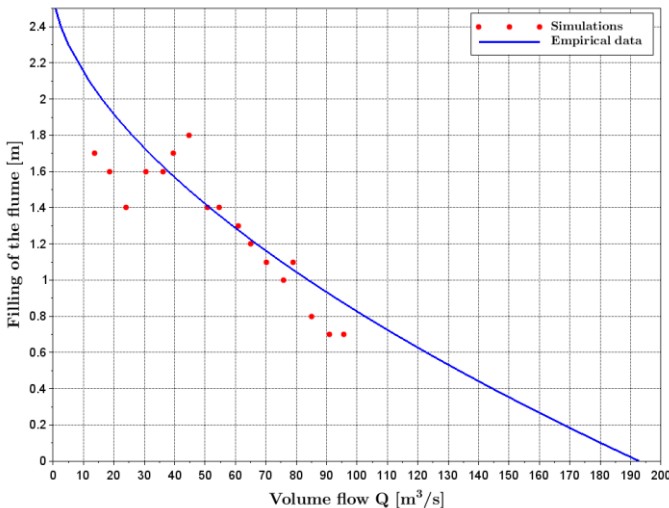

**Figure 4.** Comparison of empirical data (red) with the model curve (blue) based on Formula (4).

Empirical data should be analyzed critically too. They come from the period between September 2015 and October 2017, from the summer and autumn months. This is important because, in the adopted model, a critical role is played by the adopted roughness coefficient n, the value of which changes with the growth of the bottom river trough vegetation and siltation of the flume, and these parameters may change over months and years.

For the created model of a hydroelectric power plant operation, it is necessary to adopt a curve modeling the head-flow relation with a constant trend. It should not show sudden changes in the head with increasing flow. Therefore, it is appropriate to adopt a simplified model of constant monotonicity.

*3.2. Economic Model*

To further investigate the impact of head and flow variability on the low-head hydropower station performance, the levelized cost of electricity (LCOE) and net present value (NPV) indicators were calculated. Considering the number of uncertain parameters affecting the values of such economic indicators, the Monte Carlo (MC) approach was adopted. This method enables grasping the whole spectrum of potential scenarios where the system under investigation may operate in the future. Table 2 summarizes the input data used in the MC simulation. The energy generated annually from the low-head hydropower station of the investment period was obtained from the simulations spanning the period 1971–2018. The detailed procedure is presented in Appendix A. Equations (5) and (6) are used, respectively, to calculate the LCOE and NPV indicators.

$$\text{LCOE} = \frac{I_0 + \sum_{t=1}^{m} \frac{A_t}{(1+d)^t}}{\sum_{t=1}^{m} \frac{E_t^H}{(1+d)^t}} \tag{5}$$

where: $I_0$—investment expenditure [€], $A_t$—annual total cost [€], $E_t^H$—energy generated from hydropower [kWh], d—discount rate [%].

$$\text{NPV} = \sum_{t=1}^{m} \frac{R_t}{(1+r)^t} \tag{6}$$

where: $R_t$—net cash inflows-outflows during a single period t [€], r—discount rate [%], t—number of periods.

**Table 2.** Input parameters for the Monte Carlo (MC) simulations [45–47].

| Parameter | Value | Comment |
|---|---|---|
| Lifetime/investment period | 60 years (Carlsson, 2014) | Assuming continuous maintenance works |
| Return rate/Discount rate 1 | 5.0% (Paska, 2012) | Typical for RES investment in Poland |
| Fixed operational cost (FOM) | 1.5% (Carlsson, 2014) | Of CAPEX |
| Variable operational cost (VOM) | 3% (Carlsson, 2014) | Of CAPEX |
| Capital expenditures (CAPEX) | min: 2540, mean: 5600, max 8150. All euros/kW. (Carlsson, 2014) | Simulated in MC as a triangular distribution. |
| Certificate for hydro generation | 120 euros/kWh [47] | The latest auction won by the major hydro producer in Poland |
| Number of MC runs | 10,000 | Trial and error estimation |
| Kaplan capacity | 405 kW | Own assumptions |
| Archimedes capacity | 398.4 kW | Own assumptions |

1 regional discount rate. For simplicity, the return rate is assumed to be an equal discount rate.

## 4. Results

### 4.1. Estimating Energy Production

In the example discussed below, 8 Archimedes hydro units were selected for comparison, based on data provided by one of the manufacturers of this type of solution in Central Europe (GESS-CZ, SRO, [48]). A single hydro unit's nominal flow rate is 3.75 m³/s, with overload possibility up to 4.5 m³/s. The minimum operating head, which is the critical parameter of the comparison, is 1.2 m. A single hydro unit has a rated power of 49.8 kW.

The above-mentioned hydro units were compared with three tubular Kaplan turbines with a horizontal rotor. The data obtained from the manufacturer indicate a rated flow of 11 m³/s for a single turbine, with no overload possibility. This technology has a minimum head of 0.9 m. The rated power of a single hydro unit is 135 kW.

Based on the fact that the maximum theoretical power for the tested watercourse in the specific location is approx. 665 kW, the model presented in the article investigated the dependence of annual electricity production on the number of hydro sets installed. The magnitude of the obtained rated power was determined on the basis of the optimal use of the theoretical power of the watercourse. This was confirmed in the further part of the research and illustrated in the diagrams of annual energy production depending on the number of hydro sets installed. Based on model calculations, it was assumed that the installed power for the Archimedes screw is 398.4 kW. For comparison, it is 405 kW for Kaplan turbines.

Data from three representative years were selected for detailed analysis. The year 2006 was taken as an example of a year with a lower average annual flow, 2010 was chosen as the average reference year, and 1999 was taken as a year with a higher average flow. In Figure 5, where the average annual flow over the years 1971–2018 is presented, the years mentioned above are marked in green.

Figure 6 shows that the annual electricity production in particular years is compared with the minimum head and without it. Without considering the minimum head, the analyzed power plant operates in the full range of flows occurring at the barrage, i.e., 13.2–149 m³/s. The minimum head for the analysis changes the power plant operation scope depending on the tested technology. The element that binds the head and the flow is the previously modeled head-flow characteristic, which allows for the read of the head occurring on the tested barrage with the known flow or vice versa. The particular value of the minimum head is a parameter appropriate for a given type of hydro unit provided by the manufacturer. It refers to the lowest possible gradient at which the hydro unit can work for a long time. Therefore, each time the value of the current head is reduced below the value adopted by the manufacturer, the hydro unit must be stopped.

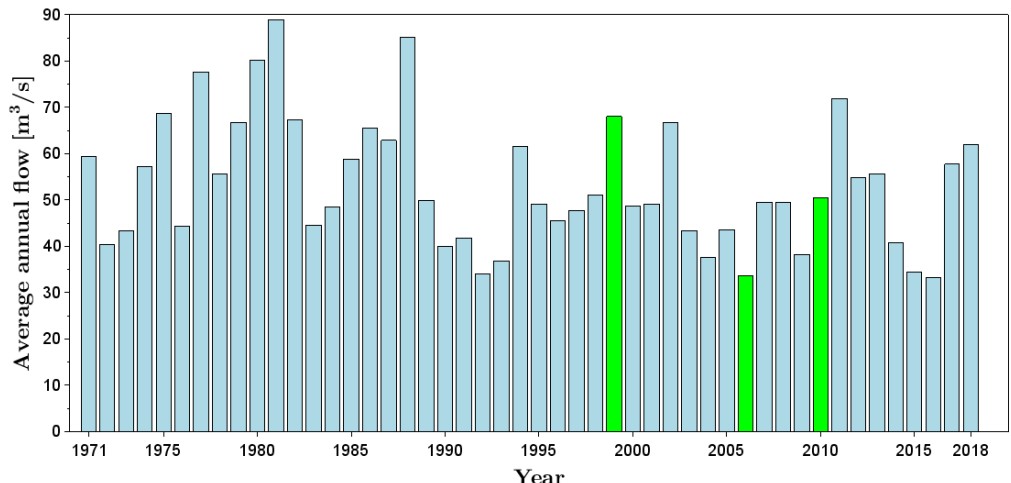

**Figure 5.** Summary of the average annual flow in 1971–2018 (the years analyzed in detail are marked in green).

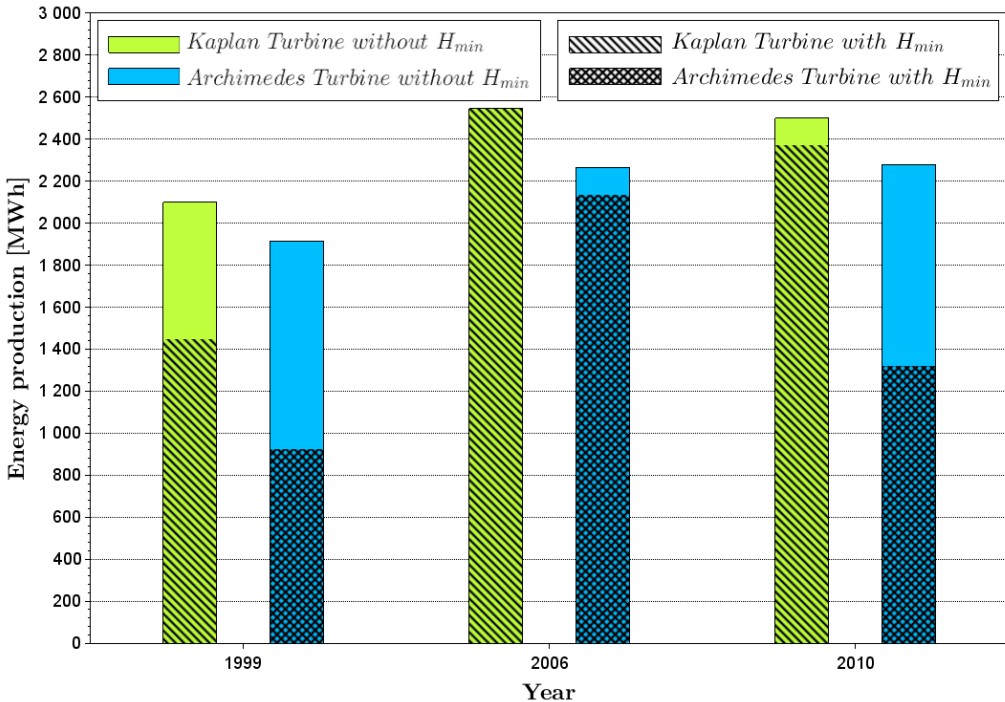

**Figure 6.** Summary of annual electricity production. Green refers to the Kaplan turbine and blue to Archimedes turbine, taking into account the minimum head. Both types of turbines are marked with hatching for calculations, excluding the minimum head impact.

The data for the three selected years presented in Figure 6 should be compared in a broader view. Figures 7 and 8 show that changes in the annual energy production are observed, which are not proportional to the average annual flow increases. The annual average flow in 2010 was 50.5% higher than in 2006, while the energy yield was lower for Kaplan and Archimedes turbines by 6.8% and 38.2%, respectively. For 1999, the observed falls in production are significant. They are equal to 38.9% (Kaplan) and 29.9% (Archimedes) in comparison to 2010, with a significant 34.8% increase in the average flow.

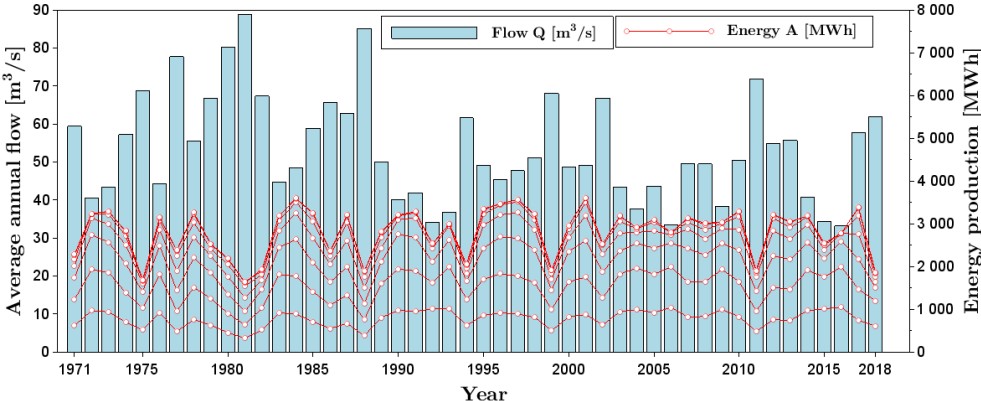

**Figure 7.** Annual electricity production (red) depending on the number of installed Kaplan turbines compared to the average annual flow (blue). The lowest red line represents one hydro unit, and the highest—eight hydro units.

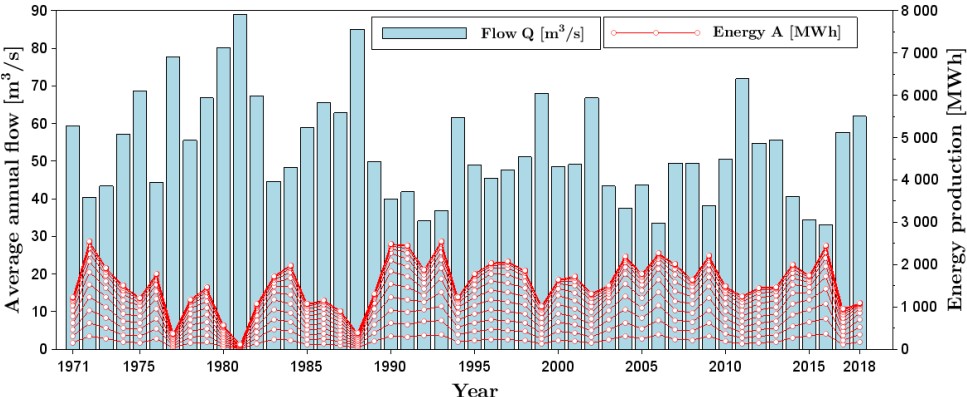

**Figure 8.** Annual electricity production (red) depends on the number of Archimedes turbines installed against the average annual flow (blue). The lowest red line represents one hydro unit and the highest—fourteen units.

Comparing Figures 7 and 8, it is possible to notice a much lower annual energy yield with the use of the Archimedes screw technology compared to Kaplan turbines. There are difficulties in using high flows in this technology.

Figures 7 and 8 show that production increase slows down after the 8th Archimedes turbine with an installed rated flow of 36 m$^3$/s and the 3rd Kaplan turbine, which corresponds to an installed rated flow of 33 m$^3$/s. The higher of these values is about 0.7 SSQ (mean flow from average annual flow) and 1.4 SNQ (average low flow). A nominal head is determined.

To explain the differences as mentioned above, Figures 9–11 are presented. They consist of several sub-charts. The first sub-chart presents the power plant's actual power and the theoretical power of the watercourse during the year. It results from, among other things, the ordered flow through the weir, i.e., the entire stream of water in the watercourse, which is shown below. Moreover, the flow through the power plant is presented, which depends on the number of running hydro units, never reaching the full water use in the watercourse. This is due to the necessity to ensure an environmental flow, the value of which is specified in the water permit and is related to the minimum environmental requirements [49]. For the considered barrage, the environmental flow is 5 m$^3$/s, maintained through, among other things, a fish pass and leaks in the weir. The last sub-chart presents the number of running turbines depending on the current head in the barrage. The discussed sub-graphs form a coherent whole allowing for analyzing the barrage's power plant operation and, in a broader sense, a comparison of this work, in years, with different average flows.

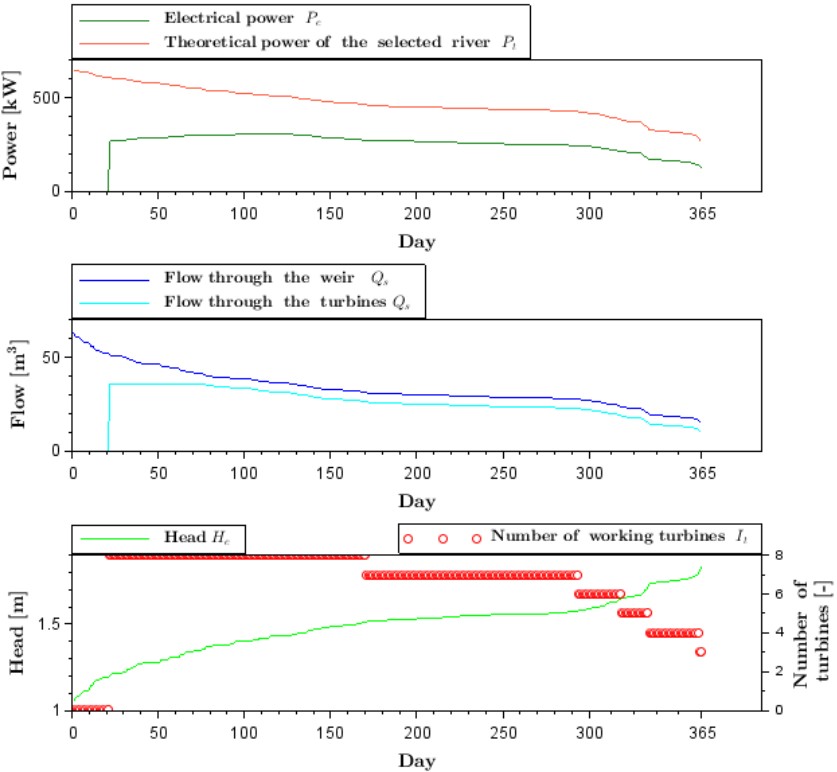

**Figure 9.** Selected operating parameters of Archimedes turbines with an average low flow below mean states (SSQ) (2006).

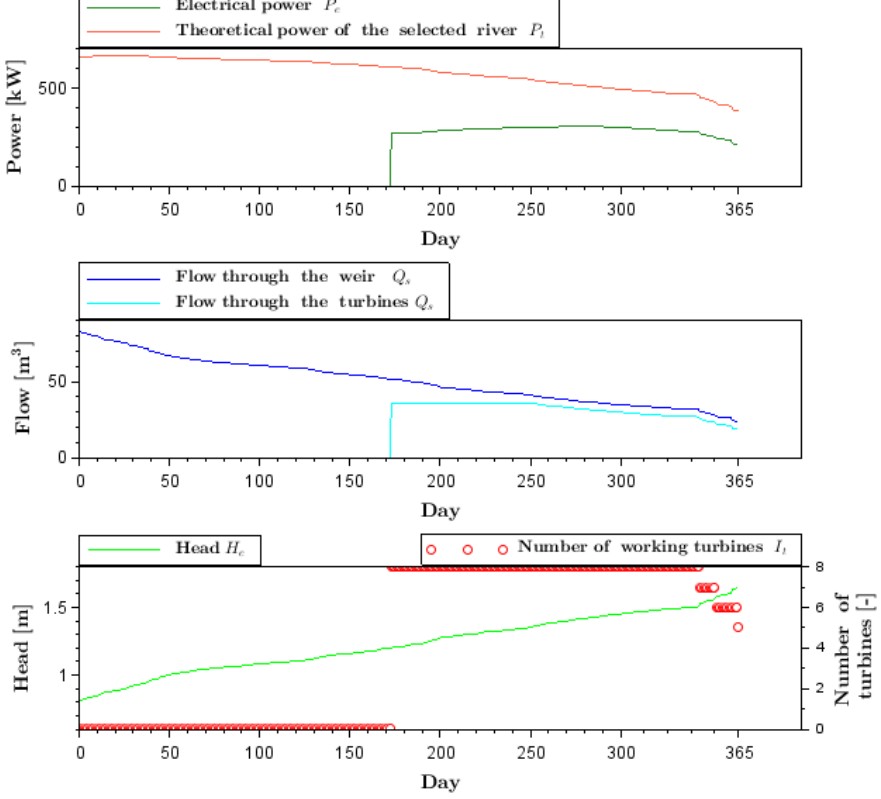

**Figure 10.** Selected operating parameters of Archimedes turbines with an average flow close to SSQ (2010).

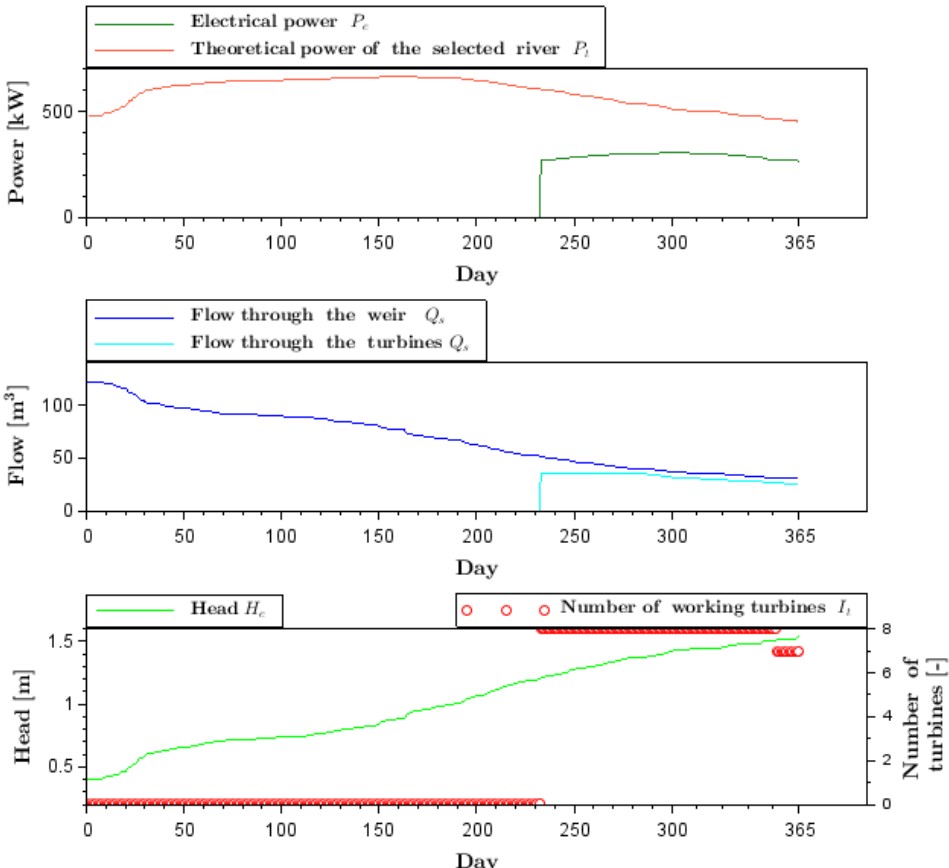

**Figure 11.** Selected operating parameters of Archimedes turbines with an average flow above SSQ (1999).

As regards the number of days when turbines were out of service, the operating parameters of Archimedes turbines in 2006 are the most favorable among those in the analyzed period. Each subsequent year, where the average annual flow was higher, the number of shutdowns was higher too. It is highly noticeable in 1999 when the number of days with hydro units working is lower than the number of forced shutdowns.

There is a difference in the minimum head for both technologies, enabling the operation of turbines, as noted at the beginning of the chapter. This results in a significantly different utilization of the theoretical energy of the watercourse. The utilization time of the Archimedes turbine installed capacity in 1999 is 1934.4 h while it is 3581.8 h for the Kaplan turbine. The value of this parameter is 85.2% higher in favor of the Kaplan turbine. It should be noted that the head value at which the hydro unit was stopped was 1.2 m for Archimedes and 0.9 m for Kaplan.

Figure 12 shows the average multiyear energy production for three selected years, depending on the number of installed turbines. An interesting and distinctive aspect is the lack of an increase in energy production if the number of Archimedes turbines exceeds twelve, which corresponds to the installed rated flow value of 54 m$^3$/s. Taking into account the environmental flow of 5 m$^3$/s, it gives a flow of 59 m$^3$/s. This value is close to the flow at the minimum head.

An analogous phenomenon occurs for Kaplan turbines, as shown in Figure 13. In this case, if the number of installed turbines is higher than six, the increase in electricity production is minimal, i.e., for a flow of 71 m$^3$/s, taking into account the environmental flow. This value is also close to the flow with the minimum head for this technological solution.

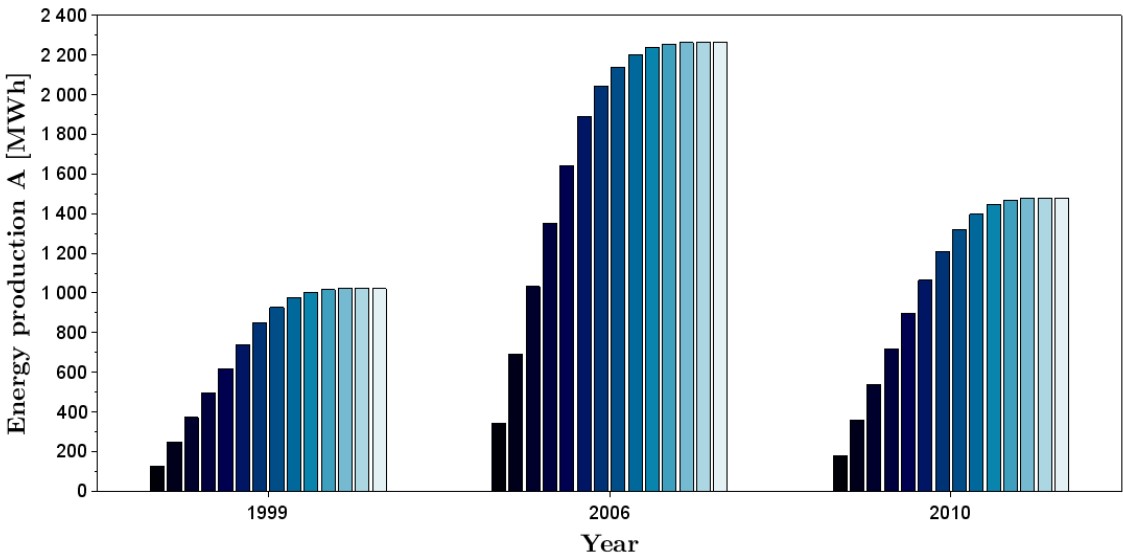

**Figure 12.** Average long-term production of energy generated by Archimedes turbines with a variable number of installed hydro sets. The leftmost bar represents one hydro unit and the rightmost—fourteen hydro units.

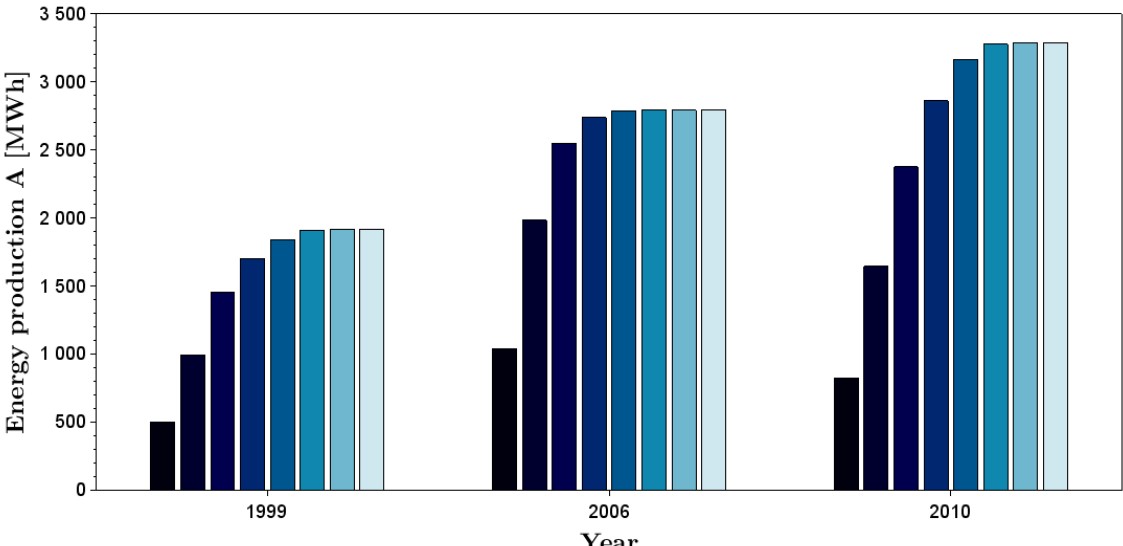

**Figure 13.** Average long-term production of energy generated by Kaplan turbines with a variable number of installed hydro sets. The leftmost bar represents one hydro unit and the rightmost eight hydro units.

*4.2. Estimating Cost–Efficiency*

Figure 14 shows the economic analysis results according to the methodology described in Section 3.2.

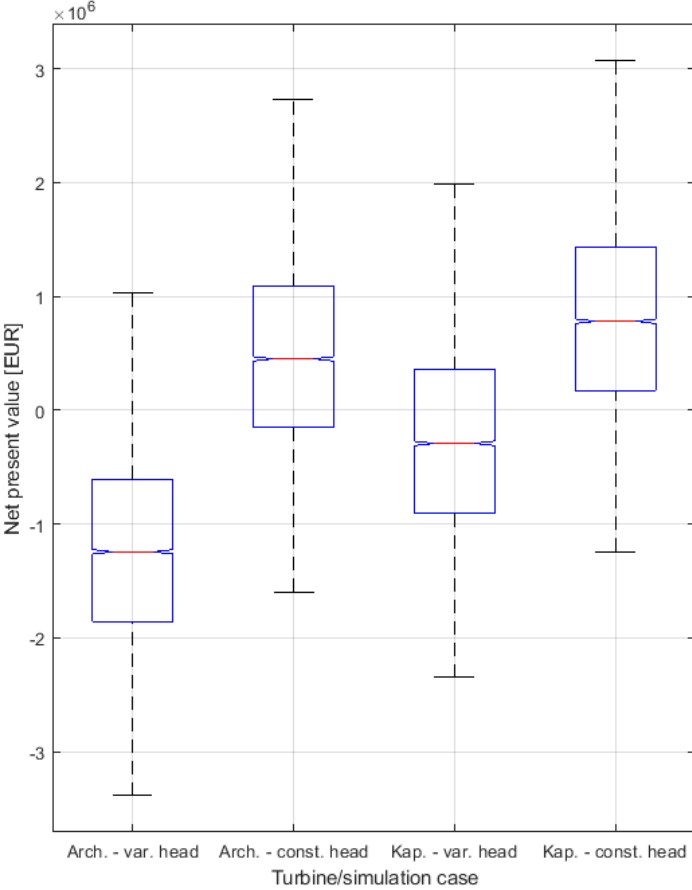

**Figure 14.** Net present value for investment in two different hydro turbines considering two simulation methods.

Considering the NPV criterion, large variances are noticed in the profitability of investments. As shown in Figure 14, assuming that the minimum head is not taken into account, both the Archimedes and Kaplan turbines are economically justified in the vast majority of cases. For the Kaplan turbine technology, the phenomenon of a negative NPV criterion (unprofitable investment) is sporadic. Considering the critical factor in the analysis, which is the hydro unit's minimum head parameter, it leads to entirely different results. Investments in a hydropower plant based on the Archimedes turbine technology are unprofitable in virtually every input variable configuration in the models. However, in the case of hydropower plants with Kaplan technology, there are some cases where the investment may be profitable. Nevertheless, the NPV indicator clearly shows that an investment in a hydroelectric power plant of such a capacity, for the assumed input parameters, is unprofitable.

Considering the occurring parameter of the hydro unit's minimum head and its impact on the electricity yield by a given hydro power plant or a turbine enables a more accurate estimation of the levelized cost of electricity (LCOE). It is a commonly accepted criterion for comparing different sources of electricity. As shown by the results presented in Figure 15, the unit cost of electricity from hydropower plants working with Archimedes and Kaplan turbines (for calculations that did not consider the minimum head) is, consecutively, 175 Euro/MWh (Archimedes), and approx. 130 Euro/MWh (Kaplan). In Poland, the current electricity price for household consumers is 137 Euro/MWh (including taxes) [50].

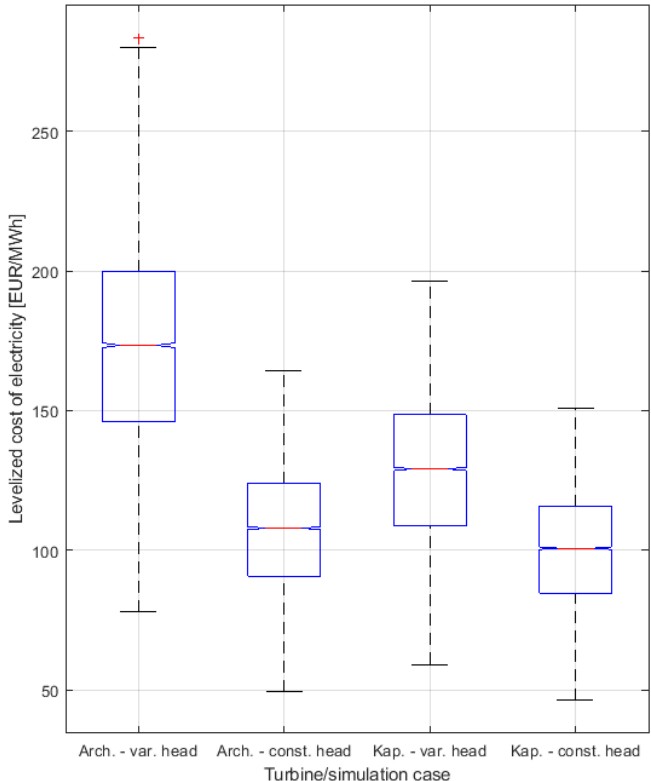

**Figure 15.** Levelized cost of electricity for two different turbines and simulation approaches.

It should also be borne in mind that the use of renewable energy resources based on watercourse brings a number of benefits, particularly the introduction of very low-emission electricity (in terms of $CO_2$) into the power system and a positive effect on low retention. These benefits are difficult to quantify and express in monetary values; therefore, they are not included in the LCOE or NPV analyses.

## 5. Discussion

The parameters on the barrage, i.e., a relatively high flow and nominal head below 1.7 m, pose an enormous challenge to the effective use of the theoretical watercourse power. Such conditions are the reality of the Polish hydropower landscape. For natural reasons, the locations with the most favorable hydrological conditions were utilized first. That is why the locations that remain under development are a great technological challenge.

Low-head locations are plentiful in Poland and, taking into account the aforementioned phenomena, it causes much pressure on hydro-set manufacturers to improve low-head solutions. The main desirable features are the ability to use increasingly lower heads with non-increasing unit costs and simultaneously retaining the required maintenance-free operation. One of the most popular small-scale hydropower development directions is tubular turbines with Kaplan rotors and Archimedes screws.

Producers of the technologies, as mentioned above, compete vigorously for the customer who needs to choose the most promising technology given the limited information available—looking more at a given technology than just the performance characteristics could help in making the right choice. The first step showing the essence of taking into account the minimum head is its impact on the annual electricity production (Figure 6). The figure compares the annual energy production results for both technologies, showing the significant impact of the minimum head on this value. This aspect has been previously omitted in research performed by most of the authors cited in the introduction to this study.

The minimum head value also significantly influences the obtained energy yields as the number of installed hydro sets increases (Figure 12, Figure 13). We can observe the negligible sense of investing

in additional hydro units above a particular value of the installed rated flow. The first of these limits is the flow that occurs at the minimum head. Exceeding the installed flow by this value will not increase the energy produced during the year. This is due to the need to stop the operation of hydro sets above that flow.

The phenomenon, as mentioned above, is illustrated in Figure 9 to Figure 11. The relationship between flow increase, head decrease on the barrage, and the number of operating hydro sets is clearly visible. For the sake of simplification in the model, the head at the weir and the power plant stand was assumed as one identical value. The modeled power plant works with water discharge into the power channel in the considered location, connecting to the river mainstream several hundred meters below the dam. It is possible that the operation of the hydro sets with the head close to the minimum parameters could be kept thanks to the limitation of the flow through the power plant and thus switching off other hydro sets. However, these considerations require further simulations and, if possible, real tests.

Another assumption is that the flow in a watercourse is uniform, while in reality, it should be slowly variable. The results of the studied model were compared with the empirical data and considered sufficiently accurate. The number of running hydro sets was also determined based on their rated flow ranges to force their operation in the optimal range of the efficiency characteristics. Furthermore, no online algorithm was performed to search for the maximum efficiency point. It should also be borne in mind that a hydropower plant's construction with an existing barrage may change the flow drop characteristics.

Earlier in this paper, Figures 12 and 13 outline the issue of determining the installed flow of low-head power plants. It seems that these values could depend on a particular multiplicity of the average low flow (SNQ), i.e., the parameter at which the nominal head occurs, and of the average flow from the mean states (SSQ). The former flow value would be the lower limit of the proposed range, and the latter would be the upper limit. This issue also requires further research, taking into account a more comprehensive range of representative river types and barrage head characteristics occurring in these watercourses.

## 6. Conclusions

The main topic of the work is to investigate the influence of the minimum head parameter on the operations of low-head hydropower plants. By modeling and analyzing the SHP at the Drawski Młyn water barrage, it has been shown that this parameter influences electricity production. This effect is more distinct for the Archimedes screw technology than for the Kaplan turbine. The decrease in energy production associated with the hydro unit minimum head parameter for Kaplan is from 0% to 30%, and from 6% to 52% for Archimedes turbines, for the years considered in Figure 6. Then, there are significant discrepancies in the economic analysis results for both types of hydro sets. At the same time, it should be pointed out that the recommended installed flow rate for low-head locations should be specified in detail because the criteria adopted so far seem to be inadequate. The rapid development of low-head technologies should be observed, as they may be devoid of the indicated shortcomings in the future. The development should be aimed at improving the minimum head parameter.

**Author Contributions:** Conceptualization, J.J., B.C. and A.G.; methodology, J.J. and A.G.; software, A.G. and Ł.K.; validation, J.J.; formal analysis, R.W.; investigation, J.J. and R.W.; resources, A.G. and Ł.K.; data curation, A.G. and Ł.K.; writing—original draft preparation, A.G. and Ł.K.; writing—review and editing, B.C., J.J. and D.Z.; visualization, D.Z.; supervision, R.W.; project administration, B.C.; funding acquisition, B.C. All authors have read and agreed to the published version of the manuscript.

**Funding:** This research was funded by the Ministry of Science and Higher Education, grant number 0711/SBAD/4468.

**Acknowledgments:** We thank Egidijus Kasiulis and Alban Kuriqi for their help and meaningful comments, which they have provided us during the revision of our paper.

**Conflicts of Interest:** The authors declare no conflict of interest.

## Abbreviations

CAPEX       capital expenditures
FOM         fixed operational cost
LCOE        levelized cost of electricity
MC          Monte Carlo simulation
NPV         net present value
SHP         small-scale hydropower
SNQ         average low flow
SSQ         average flow from the average flows
SWR         average high flow
TSO         transmission system operator
VOM         variable operational cost

## Appendix A

The simulated energy generation from hydro turbines over 1974–2018 did not fit any standard statistical distribution. Considering the above, the procedure applied was to generate random energy generation values while maintaining the simulated time series of statistical parameters. The first step involved creating a histogram (Figure A1), which would show how the energy was generated, on an annual basis, over that period. In this case, the bins' width was selected as 100 MWh, but it can be adjusted according to the desired granularity.

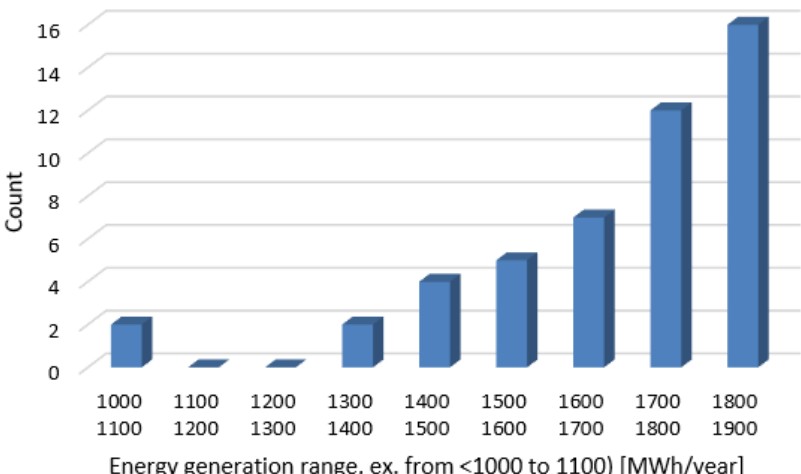

**Figure A1.** Histogram of energy generation over the period 1974–2018.

The next step in generating random values from their historical distribution involved calculating the probability of energy generation within each bin range. The calculated probabilities are presented in Table A1.

**Table A1.** Probability of energy generation from a given range.

| Range [MWh] | % Probability |
| --- | --- |
| (1000–1100> | 4.17% |
| (1100–1200> | 0.00% |
| (1200–1300> | 0.00% |
| (1300–1400> | 4.17% |
| (1400–1500> | 8.33% |
| (1500–1600> | 10.42% |
| (1600–1700> | 14.58% |
| (1700–1800> | 25.00% |
| (1800–1900> | 33.33% |

In the third step, the cumulative probability is calculated (Table A2).

**Table A2.** Cumulative probability.

| Range [MWh] | Cumulative probability |
|---|---|
| (1000–1100> | 4.17% |
| (1100–1200> | 4.17% |
| (1200–1300> | 4.17% |
| (1300–1400> | 8.33% |
| (1400–1500> | 16.67% |
| (1500–1600> | 27.08% |
| (1600–1700> | 41.67% |
| (1700–1800> | 66.67% |
| (1800–1900> | 100.00% |

Finally, the energy generation time series can be replicated by applying the following approach:

Step 1. Generate random numbers p(1) from a uniform distribution ranging from 0 to 1;

Step 2. For the generated number, check if it is less than the cumulative probability for the first range (1000–1100 MWh). For example, if that is the case, the random number p(1) was 0.02, proceed to Step 3; if not, proceed to Step 4;

Step 3. For the range for which the p number was smaller than the cumulative probability, generate another number from a uniform distribution in which minimum and maximum values are, as in the range boundaries, for the case that would mean generating a number from a uniform distribution ranging from 1000 to 1100 MWh. The generated number is then the first simulated energy generation value. Proceed to Step 1 to generate the following energy generation values;

Step 4. In this step, move to the next range and its cumulative probability values. If the cumulative probability values are similar for two or more ranges, remove the subsequent ones and keep only the first one. In Table A2, the range 1000–1100 would be kept only, while 1100–1200, 1200–1300 would have to be removed. Select the first range for which the p number is smaller than its cumulative probability value. For example, for p equal to 0.25, it would be range 1500–1600. For p equal to 0.6, it would be 1700–1800. Repeat this step until the p-value is assigned to the range. If found, proceed to step 1;

Step 5. The simulation ends when the desired number of energy generation numbers has been reached.

To assess the usefulness of the proposed approach, we have generated 3000 energy generation numbers whose distribution should follow the historical data obtained from simulations over the period 1974–2018. The chart in Figure A2 compares the simulation results from historical data and after applying the procedure proposed above. The proposed method enables a perfect fit with the original distribution.

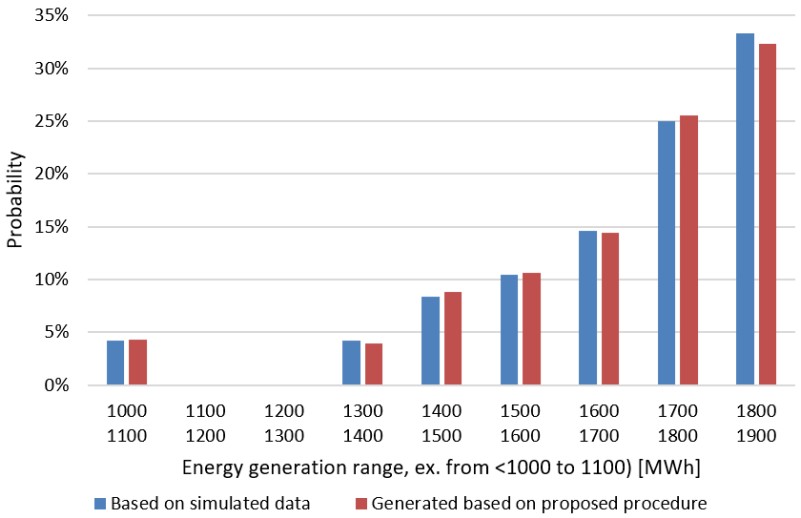

**Figure A2.** The probability of energy distribution from different ranges is based on historical data simulations and recreating the distribution based on the proposed method.

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
