# Peer review of "Impact of the Minimum Head on Low-Head Hydropower Plants Energy Production and Profitability"

_energies, doi:10.3390/en13246728_

Round 1
Reviewer 1 Report
I have read carefully the article titled: "Impact of minimal head on low-head hydropower plants energy production and profitability", id: energies-1000134. The article discusses the economic analysis of two turbine designs (Archimedes and Kaplan) to a low head application in a particular location in Poland. The article is complete, in the sense that it includes all necessary aspects for calculations and it is relatively well strutured and written. My main concern is on the novelty of the present work, in the sense that the procedures discussed are well known and applied in all prospective applications of micro-hydropower. With that, my intention is not to reduce the effort of the authors; I perfectly understand the incentive of the analysis, however it is not something novel. I would recommend to the authors to highlight strong points of their investigation and how these advance the know-how in microhydropower applications.
Minor comments:
- The quality of figures 3 and 4 has to be improved.
- Lines 139-143: the reference to chapters is inappropriate. Prefer "sections" instead.
- The language can be improved (e.g. line 341 "In order explain"), especially when referecing work of others (e.g. line 155: "others [34] give", line 162: "The researches", etc ). Also, in line 317 a gap is missing ("Figure6").
Author Response
Reply in pdf attachment.
Reviewer 2 Report
The paper presents the impact of the minimal head of the hydro sets on the annual electricity production of small hydropower plants at low-head locations for two types of water turbines both Archimedes and Kaplan. The Author's attempt is to develop a mathematical model to simulate energy.
In the Introduction literature review, each citation should be done individually for a single reference. Clubbing of more than one referred article by one single statement for citation as it is done in several cases should be avoided otherwise it would be inferred that citations are done only for the formality without having focused and precise relevance. Write up for the research gap should be in a separate single paragraph to create the appropriate prelude for the motivation of the work.
Write up for the research gap should be in a separate single paragraph to create the appropriate prelude for the motivation of the work. Actually at the end of the introduction that should summarise condensed knowledge following with problem definition last paragraph is not acceptable. It should be rewritten according to aforementioned.
Actually the next part of the manuscript remains more engineering work than scientific. Compares two approaches that are apparently already used and/or already in the market. Therefore, It is not able to consider the paper as a manuscript delivering some novelty in the given field.
Author Response
Reply in pdf attachment.
Round 2
Reviewer 2 Report
In my opinion the authors precisely updated manuscript incorporating all remarks listed in the review report. Manuscript now is much more consistent and can be considered for publication in the journal. Good luck.